# Novel Probiotic Bacterium *Rouxiella badensis* subsp. *acadiensis* (Canan SV-53) Modulates Gut Immunity through Epigenetic Mechanisms

**DOI:** 10.3390/microorganisms11102456

**Published:** 2023-09-29

**Authors:** Roghayeh Shahbazi, Hamed Yasavoli-Sharahi, Jean-François Mallet, Farzaneh Sharifzad, Nawal Alsadi, Cyrille Cuenin, Vincent Cahais, Felicia Fei-Lei Chung, Zdenko Herceg, Chantal Matar

**Affiliations:** 1Cellular and Molecular Medicine Department, Faculty of Medicine, University of Ottawa, Ottawa, ON K1H 8M5, Canada; 2Department of Urology, Feinberg School of Medicine, Northwestern University, Chicago, IL 60611, USA; 3Epigenomics and Mechanisms Branch, International Agency for Research on Cancer (IARC), 25 Av. Tony Garnier, 69007 Lyon, France; 4Department of Medical Sciences, School of Medical and Life Sciences, Sunway University, Jalan Universiti, Bandar Sunway, Subang Jaya 47500, Selangor, Malaysia; 5School of Nutrition, Faculty of Health Sciences, University of Ottawa, Ottawa, ON K1H 8M5, Canada

**Keywords:** gut immunity, gut microbiome, probiotic SV-53, prebiotic, Th17 cell, miRNA, DNA methylation

## Abstract

Gut immune system homeostasis is crucial to overall host health. Immune disturbance at the gut level may lead to systemic and distant sites’ immune dysfunction. Probiotics and prebiotics consumption have been shown to improve gut microbiota composition and function and enhance gut immunity. In the current study, the immunomodulatory and anti-inflammatory effects of viable and heat-inactivated forms of the novel probiotic bacterium *Rouxiella badensis* subsp. *acadiensis* (Canan SV-53), as well as the prebiotic protocatechuic acid (PCA) derived from the fermentation of blueberry juice by SV-53, were examined. To this end, female Balb/c mice received probiotic (viable or heat-inactivated), prebiotic, or a mixture of viable probiotic and prebiotic in drinking water for three weeks. To better decipher the immunomodulatory effects of biotics intake, gut microbiota, gut mucosal immunity, T helper-17 (Th17) cell-related cytokines, and epigenetic modulation of Th17 cells were studied. In mice receiving viable SV-53 and PCA, a significant increase was noted in serum IgA levels and the number of IgA-producing B cells in the ileum. A significant reduction was observed in the concentrations of proinflammatory cytokines, including interleukin (IL)-17A, IL-6, and IL-23, and expression of two proinflammatory miRNAs, miR-223 and miR425, in treated groups. In addition, heat-inactivated SV-53 exerted immunomodulatory properties by elevating the IgA concentration in the serum and reducing IL-6 and IL-23 levels in the ileum. DNA methylation analysis revealed the role of heat-inactivated SV-53 in the epigenetic regulation of genes related to Th17 and IL-17 production and function, including *Il6*, *Il17rc*, *Il9*, *Il11*, *Akt1*, *Ikbkg*, *Sgk1*, *Cblb*, and *Smad4*. Taken together, these findings may reflect the potential role of the novel probiotic bacterium SV-53 and prebiotic PCA in improving gut immunity and homeostasis. Further studies are required to ascertain the beneficial effects of this novel bacterium in the inflammatory state.

## 1. Introduction

The largest immune system compartment in the body belongs to the gut [1]. Paneth cells, Goblet cells, and gut-associated lymphoid tissue (GALT) contribute to the gut immune system function [2,3,4]. Paneth cells produce antimicrobial peptides such as lysozyme and β-defensins, while Goblet cells produce mucus [2,3,4]. Peyer’s patches, found in GALT, consist of B cell and T cell zones and are the main source of small intestine immunoglobulin A (IgA) plasmablasts [3,4]. The mucus layer, antimicrobial peptides, and IgA play crucial roles in gut barrier integrity and mucosal immunity [5].

Adaptive immune cells, including regulatory T cells (Treg) and T helper-17 (Th17), play a crucial role in maintaining gut immune system homeostasis. Signals from T cell receptors and cytokines direct T cells balance in the gut [6]. Th17 cells are a distinct CD4^+^ T-helper subtype characterized by interleukin (IL)-17 production that promotes inflammatory responses [7]. Differentiation of naïve CD4^+^ T cells into Th17 cells is regulated by the retinoic acid receptor-related orphan receptor-gamma-t (RORγt) transcriptional factor [8]. Transforming growth factor-β (TGF-β) and IL-6 are necessary for the early-stage differentiation of Th17 cells, while IL-23 plays a critical role in expanding Th17 cells and enhancing their pathogenic functions [9,10]. IL-10 inhibits Th17 cells development and immune responses [11]. Over-production of Th17 cells is associated with autoimmunity and inflammatory diseases such as inflammatory bowel disease (IBD) [6,12].

In addition to the role of cytokine milieu and transcription factors in the differentiation of naïve CD4^+^ T cells [13], the presence of Th17 in the small intestine is influenced by gut microbiota and microRNAs (miRNAs) profile [14,15,16]. Gut miRNAs play pivotal roles in both innate and adaptive immunity, directing various physiological and immunological processes in the intestine. Specific miRNAs contribute to the proliferation of intestinal epithelial stem cells, epithelial regeneration, differentiation of Paneth and Goblet cells, and the balance between Th17 and Treg cells [16]. Additionally, epigenetic mechanisms, including DNA methylation and posttranslational histone modifications, contribute to T cell generation [15]. 

Probiotics and prebiotics have been shown to enhance gut immunity by beneficially affecting gut microbial communities, maintaining the gut epithelial barrier, inhibiting pathogens’ growth, and modulating innate and adaptive immune responses [17,18]. Probiotics have also been found to modulate the miRNAs expression involved in gut immunity [19]. In addition, heat-inactivated probiotics, their fractions, and purified components have been demonstrated to confer health benefits by protecting against pathogens and enhancing intestinal barrier function [20]. Probiotics and prebiotics may also enhance gut immunity by reducing proinflammatory cytokines such as IL-6 and IL-1β and increasing anti-inflammatory cytokines such as IL-10 [21]. Additionally, probiotics, prebiotics, and their metabolites have been shown to boost gut immunity by modulating epigenetic mechanisms, including DNA methylation and histone modification [22,23,24]. 

We have previously described the probiotic characteristics of a novel Gram-negative probiotic, *Rouxiella badensis* subsp. *acadiensis* (Canen SV-53), referred to as SV-53. This probiotic microorganism, which was isolated from the natural microflora of lowbush blueberry [25], reinforces intestinal homeostasis by increasing the number of Paneth cells and production of the antimicrobial peptide α-defensin [26]. In the current study, the immunomodulatory properties of the live and heat-inactivated SV-53 and prebiotic protocatechuic acid (PCA) derived from fermented blueberry juice by SV-53, were studied in female Balb/c mice by analyzing the effects of biotics intake on gut microbiota, gut mucosal immunity, and selected cytokines and miRNAs involved in Th17 cells differentiation and function. We also aimed to study whether SV-53 exerts potential immunomodulatory activities through modulating epigenetic mechanisms by analyzing its effect on the DNA methylation status of genes related to Th17 cell function. 

## 2. Materials and Methods

### 2.1. Animals

Eight-week-old female Balb/c mice (Charles River, Montreal, QC, Canada) were used in the current study. Three mice were housed together in plastic cages in a controlled atmosphere (temperature 22 ± 2 °C; humidity 55 ± 2%) with a 12 h light/dark cycle. During the study, all groups received a conventional balanced diet ad libitum. Mice were maintained and treated according to the guidelines of the Canadian Council on Animal Care. The protocol (HSe-3191) was approved by the Animal Care Committee of the University of Ottawa.

### 2.2. Probiotic and Prebiotic Solution Preparation

In the current study, we used the research bank cultures of the bacterium prepared by the National Research Council of Canada. The bacteria cultures were maintained in a tryptic soy broth (TSB) culture medium (Difco Laboratories, Detroit, MI, USA) supplemented with 30% (*v*/*v*) glycerol at −80 °C. 

To prepare the probiotic solution, the bacteria were cultured in TSB for 17 h at room temperature. Then, the bacteria culture was centrifuged at 5000 rpm for 10 min. The bacterial pellet was washed three times in sterile PBS (Sigma, Saint Louis, MO, USA) and resuspended in 5 mL of sterile 10% (wt/vol) non-fat milk. Bacterial suspensions were diluted 1:30 in water and administered ad libitum to the mice at a final concentration of 5 × 10^7^ CFU/mL. To prepare heat-inactivated bacteria, the same procedure was applied, and 5 × 10^7^ CFU/mL bacterial preparation in the sterile water was heated at 70 °C for 30 min. The heat-inactivated preparations were kept at −80 °C to feed the mice. An average of 2 mL of probiotic solution was consumed daily by each mouse; therefore, the mice approximately received a daily dose of 1 × 10^8^ CFU of the live and heat-inactivated bacterium. 

To prepare the prebiotic solution, 100 mg/kg BW of PCA (3,4-dihydroxybenzoic acid) was dissolved in sterile water. Then, the pH of the prebiotic solution was adjusted to 7.4 using a pH meter. The fresh mixture was prepared twice a week and kept at 4 °C.

### 2.3. Study Design

#### 2.3.1. Probiotic-Prebiotic Experiment

Mice (n = 24) were categorized into four groups: 1—control group; receiving sterile water, 2—probiotic group; receiving probiotic SV-53 (5 × 10^7^ CFU/mL) in sterile drinking water, 3—prebiotic group; receiving 100 mg/kg prebiotic PCA in sterile drinking water, 4—probiotic + prebiotic group; receiving a mixture of SV-53 and PCA in sterile drinking water. The duration of the nutritional intervention was three weeks. Afterward, mice were sacrificed, and the required samples, including blood, feces, and ileum tissues, were collected and stored at −80 °C to conduct corresponding experiments. 

#### 2.3.2. Heat-Inactivated Probiotic Experiment

Mice (n = 18) were divided into three groups: 1—control; receiving sterile drinking water, 2—probiotic group; receiving live SV-53 (5 × 10^7^ CFU/mL) in sterile drinking water, and 3—heat-inactivated probiotic group; receiving heat-inactivated SV-53 (5 × 10^7^ CFU/mL) in sterile drinking water. At the end of three weeks of nutritional intervention, mice were sacrificed, and the required samples were collected and kept at −80 °C until further experiments were performed.

### 2.4. Histological Sections Preparation

The ileum tissues of mice were removed, washed with ice-cold PBS, and small sections of the ileum were collected and fixed in a 4% paraformaldehyde solution for 48 h. Subsequently, fixed tissues were dehydrated in increasing alcohol concentrations, cleared in xylene, and embedded in paraffin using conventional methods. Histological sections of 4 µm were prepared from paraffin blocks using a rotational microtome (Leica RM2255 Automated Microtome). The processing, embedding, and preparation of histological sections were performed by the University of Ottawa Histology Core Facility.

### 2.5. Identification of IgA, IgG, IL-17A, IL-6, IL-23, and IL-10 Producing Cell Populations by Immunofluorescence

The numbers of IgA^+^, IgG^+^, and IL-17A^+^ cells on histological sections from the ileum region were determined via direct immunofluorescence. The immunofluorescence tests were performed using FITC-conjugated goat (α-chain specific) polyclonal anti-mouse IgA (Sigma-Aldrich, St. Louis, MO, USA), FITC-conjugated goat (γ-chain specific) polyclonal anti-mouse IgG (Sigma-Aldrich, St. Louis, MO, USA), and anti-Mo/Rat IL-17A (ebio17B7), (eBioscience, Thermo Fisher Scientific, Burlington, ON, Canada). The histological sections were deparaffinized in xylene and rehydrated in a graded series of ethanol from 95% to 40%. The deparaffinized histological samples were stained with appropriate antibody dilution in 1X PBS (1:100 for IgA, 1:50 for IgG, and IL-17A) for 1 h at 37 °C. Indirect immunofluorescence was used to determine the number of IL-6 and IL-10-producing cells using polyclonal anti-murine IL-10 (PeproTech, Rocky Hill, NJ, USA) and polyclonal anti-murine IL-6 (PeproTech, Rocky Hill, NJ, USA) antibodies, respectively. The deparaffinized histological slides were incubated with appropriate primary antibody dilution in 1X PBS (1:50) overnight at 4 °C and FITC-conjugated affinipure goat anti-rabbit IgG (H + L) secondary antibody (1:50 in 1X PBS) for 1 h at 37 °C in the dark. The slides were washed three times in PBS, mounted using Fluoromount (Sigma-Aldrich, St. Louis, MO, USA), and examined using a fluorescent light microscope (Evos FL Auto 2, Thermo Fisher Scientific, Bothell, WA, USA). The results were expressed as the number of positive cells (fluorescent cells) per 10 fields at 40× magnification.

### 2.6. Determination of IgA, IgG, IL-17A, IL-6, IL-23, and IL-10 Concentrations by ELISA

Blood samples were collected and centrifuged at 10,000× *g* for 2 min to separate serum. Serum samples were kept at −80 °C. Small pieces of ileum were snap-frozen and kept at −80 °C until protein extraction. To examine the cytokine levels, small parts of the ileum tissues (20–25 g) were collected in microtubes containing lysis buffer (Pierce IP Lysis, Thermo Fisher Scientific) and protease/phosphatase inhibitors (Halt Protease and Phosphatase Inhibitor Cocktail, Thermo Fisher Scientific) and homogenized using an electrical homogenizer (Bead mill 24, Fisher Scientific, USA). IgA and IgG levels were determined in serum using mouse IgA and IgG uncoated ELISA kits (Invitrogen, Vienna, Austria). The levels of IL-6, IL-10, IL-17A, and IL-23 were measured in ileum tissues using mouse-uncoated ELISA kits (Invitrogen, Vienna, Austria). All ELISA tests were performed according to the manufacturer’s instructions. Absorbance was read at 450 nm with a wavelength correction of 570 nm using a microplate reader (Bio-TEK Instruments, Winooski, VT, USA).

### 2.7. Determination of miRNA Expression by Real-Time Quantitative Reverse Transcription PCR (RT-qPCR)

Small pieces of ileum were placed in tubes containing RNAlater Stabilization Solution (Invitrogen, Carlsbad, CA, USA) for 24 h and then stored at −80 °C until RNA extraction. Total RNA from the samples was extracted using miRNeasy mini kit (Qiagen, Toronto, ON, Canada). Samples purity was verified with a NanoDrop 2000 (Thermo Scientific, Waltham, MA, USA). A reverse transcription reaction was performed to synthesize cDNA using the miRCURY LNA RT Kit (Qiagen, Toronto, ON, Canada). The expression of miR-223 and miR-425 was measured via RT-qPCR using hsa-miR-425-5p and hsa-miR-223-3p miRCURY LNA miRNA PCR assay primers (Qiagen, Toronto, ON, Canada) and miRCURY LNA SYBR Green PCR Kit (Qiagen, Toronto, ON, Canada) in a CFX 384 real-time PCR detection system (Bio-Rad, Laboratories, Hercules, CA, USA). miRNA expression was normalized to U6 as the reference gene using the U6 snRNA (hsa, mmu) miRCURY LNA miRNA PCR assay (Qiagen, Toronto, ON, Canada).

### 2.8. Gut Microbiome Analysis

For microbiome analysis, cecum contents of mice were collected in sterile microtubes and snap-frozen in liquid nitrogen before storage at −80 °C. Shallow shotgun sequencing was used for microbiome analysis. Microbiome analysis was performed by Microbiome Insights Company, Vancouver, BC. DNA was extracted using the MagAttract PowerSoil DNA KF kit (Qiagen, Toronto, ON, Canada). The quality of extracted DNA was evaluated visually through gel electrophoresis and quantified using a Qubit 3.0 fluorometer (Thermo-Fischer, Waltham, MA, USA). Libraries were prepared using an Illumina Nextera library preparation kit with an in-house protocol (Illumina, San Diego, CA, USA).

### 2.9. Methylome-Wide Profiling and Data Analysis

Approximately 15–20 g of the mice ileum samples was homogenized using an electrical homogenizer in tubes containing 500 µL cell lysis buffer and 1.5 µL proteinase K. The Gentra Puregene Tissue Kit (33 g) (Qiagen, Toronto, ON, Canada) was then employed to extract DNA from the tissues, according to the manufacturer’s instructions. The extracted DNA was quantified using Qubit 4 (Thermo Fisher Scientific, Waltham, MA, USA) and diluted with DNA rehydration solution to achieve a final concentration of 20 ng/µL, then stored at −20 °C. Methylome-wide profiling was conducted as previously described [27]. Briefly, 500 ng of extracted DNA was subjected to bisulfite conversion using the EZ DNA Methylation kit (Zymo Research, Irvine, CA, USA). Next, 250 ng bisulfite-modified DNA was analyzed using the Infinium Mouse Methylation BeadChip arrays, which allow for the simultaneous assessment of DNA methylation at more than 285,000 CpG sites (Illumina Inc., San Diego, CA, USA). Methylome-wide data were analyzed using the methylkey pipeline developed by the Epigenomics and Mechanisms Branch at the International Agency for Research on Cancer (https://github.com/IARCbioinfo/methylkey (accessed on 21 January 2023)). Briefly, raw data files were pre-processed, quality control was conducted, and normalization was performed via Noob normalization using the SeSAMe package [28]. Intergroup comparisons were conducted using linear regression analysis as implemented in the limma R package [29]. Regional analysis to identify differentially methylated regions was conducted using the DMRcate package [30].

### 2.10. Statistical Analysis

All statistical analyses, except for the microbiome analysis, were conducted using GraphPad Prism (GraphPad Software, Inc., La Jolla, CA, USA). One-way analysis of variance (ANOVA) followed by Dunnett’s multiple comparisons test was employed to compare groups. Data were considered significantly different when the *p*-value < 0.05. All values are mean ± SEM, based on at least three independent tests. Microbiome analysis was performed using Qiime2, and for comparisons of differentially abundant taxa, negative binomial models (DESEq2 R package) were utilized. Alpha diversity was calculated from taxonomic profiles using Shannon’s diversity index. Beta diversity analysis was conducted using Bray–Curtis dissimilarities, and the results were visualized using nonmetric multidimensional scaling (NMDS). A corrected/adjusted *p*-value < 0.05 was considered statistically significant. For Methylation analysis, differentially methylated genes were defined with a false discovery rate (FDR)-adjusted *p*-value < 0.05 and an absolute inter-group beta value difference of >0.05. Pathway visualization was performed using KEGG pathway enrichment analysis with Enrichr. 

## 3. Results

### 3.1. Effect of Probiotic and Prebiotic Intake on Mucosal Immunity

The effect of the nutritional intervention on mucosal immunity was assessed by measuring IgA and IgG levels in the serum, as well as the population of IgA and IgG-producing cells in the lamina propria of the ileum tissues. Feeding mice with SV-53 and SV-53 + PCA mixture significantly increased the serum IgA level (*p* < 0.05) (Figure 1A). Furthermore, the population of IgA^+^ cells exhibited a significant increase in the ileum tissues of treated groups compared to the untreated group (*p* < 0.05, *p* < 0.01, and *p* < 0.0001, respectively) (Figure 1B). However, neither the serum IgG level nor the number of IgG+ cells in the ileum of mice changed following the feeding period (Figure 1C,D). Figure 1E illustrates immunofluorescence images of histological sections stained with FITC-conjugated anti-IgA antibody.

### 3.2. Effect of Probiotic and Prebiotic Intake on the Expression of Selected Cytokines and miRNAs 

Oral administration of SV-53 and PCA for three weeks led to a significant reduction in IL-17A (*p* < 0.01, *p* < 0.05, and *p* < 0.01, respectively) (Figure 2A), IL-6 (*p* < 0.01, *p* < 0.05, and *p* < 0.01, respectively) (Figure 2B), and IL-23 (*p* < 0.01, *p* < 0.05, and *p* < 0.001, respectively) (Figure 2C) concentrations in the ileum tissues of treated mice compared to the control counterparts. The level of IL-10 increased in all treatment groups; however, statistical significance was reached only in the prebiotic group (*p* < 0.05) (Figure 2D). 

Additionally, the number of IL-17A, IL-6, and IL-10-producing cells in the lamina propria was measured through immunofluorescence. In accordance with ELISA results, the quantity of IL-17A-producing cells was significantly lower in all treatment groups than in the control (*p* < 0.01, *p* < 0.05, and *p* < 0.01, respectively) (Figure 2E). IL-6 producing cells frequency was reduced in mice fed SV-53 + PCA mixture compared to the control (*p* < 0.05) (Figure 2F), while the abundance of IL-10-producing cells increased in SV-53 and PCA-fed mice (*p* < 0.01 and *p* < 0.05, respectively) (Figure 2G). 

Furthermore, the impact of SV-53 and PCA intake on the expression of two pro-inflammatory miRNAs, miR-223 and miR-425, was analyzed. Feeding mice with SV-53, PCA, and their mixture significantly decreased the expression of miR-223 (*p* < 0.05) and miR-425 (*p* < 0.01, *p* < 0.001, and *p* < 0.001, respectively) in ileum tissues of mice compared to the control group (Figure 2H,I).

### 3.3. Effect of the Probiotic and Prebiotic Intake on the Gut Microbiome

Alpha diversity, calculated from taxonomic profiles using Shannon’s diversity index, did not show significant differences across groups (Figure 3A). Beta diversity was assayed using Bray–Curtis dissimilarities and samples were visualized using nonmetric multidimensional scaling (NMDS). Samples were very similar in general and no significant difference was observed (Figure 3B). Figure 3C displays the ten most abundant phyla in all groups. Microbiome analysis revealed the intestinal colonization of SV-53 in the mice that received SV-53 and SV-53 + PCA (*p* < 0.0001) (Figure 3D) and a significant decrease in the abundance of *Escherichia coli* (*E. coli*) in the SV-53 and SV-53 + PCA groups (*p* < 0.0001) (Figure 3E). 

### 3.4. Effect of Heat-Inactivated SV-53 Intake on Mucosal Immunity

The impact of heat-inactivated SV-53 on gut mucosal immunity was investigated in a separate experiment. Three weeks of treatment with heat-inactivated bacteria led to a significant increase in serum IgA levels (*p* < 0.05) with no significant effect on the IgA^+^ cells population in the lamina propria (Figure 4A,B, respectively). Heat-inactivated bacteria also had no significant effect on IgG levels and the population of IgG^+^ cells in lamina propria (Figure 4C,D, respectively).

### 3.5. Effect of Heat-Inactivated SV-53 Intake on the Expression of Selected Cytokines and miRNAs 

IL-17A levels in the ileum tissues of mice were insignificantly lower in the heat-inactivated SV-53 group than those in the control group (Figure 5A). However, IL-6 and IL-23 concentrations significantly decreased in the ileum tissue of mice following treatment with heat-inactivated SV-53 (*p* < 0.05 and *p* < 0.01, respectively) (Figure 5B,C). The heat-inactivated SV-53 treatment had no significant effect on the IL-10 level in the ileum of mice (Figure 5D). Administration of heat-inactivated SV-53 significantly reduced the numbers of IL-17A^+^ and IL-6^+^ cells compared to the control (*p* < 0.05) (Figure 5E,F, respectively), while it did not change the IL-10-producing cells’ abundance (Figure 5G). Additionally, the administration of heat-inactivated bacteria did not alter the expression of miR-223 and miR-425 in the ileum tissues of mice compared to the untreated group (Figure 5H,I).

### 3.6. Effect of Heat-Inactivated SV-53 on DNA Methylation Status 

Finally, the impact of heat-inactivated SV-53 on the methylation status of genes in the ileum samples of mice was analyzed. The methylation status of regions around the transcriptional start sites (TSS) plays a role in regulating gene expression [31,32], where hypomethylation and hypermethylation correlate with transcriptional activation and suppression, respectively [33].

Absolute group mean difference in beta values of >0.05 and FDR adjusted *p*-value < 0.05 were applied to identify differentially methylated probes (DMPs), which were used as a basis for regional analysis to find statistically significant differentially methylated regions (DMRs). We found significant DMRs related to different signaling pathways such as MAPK, PI3K-Akt, JAK-STAT, forkhead box O (FoxO), Th17 differentiation, and IL-17 signaling pathways. Figure 6A illustrates the top enriched pathways for genes associated with the identified DMRs. Notably, significant alterations in the methylation status of some genes regulating Th17 differentiation and function were observed in mice receiving heat-inactivated SV-53 compared to the control (Figure 6B). Significant hypermethylation of CpGs was identified within the promoters of *Il6*, IL-17 receptor c (*Il17rc*), and *Il9*, and the 1 to 5 kb region of *Il11*. In addition, the 1 to 5 kb region of *Akt1* and the promoters of inhibitor of nuclear factor kappa B kinase regulatory subunit gamma (*Ikbkg*) and serum glucocorticoid kinase 1 gene (*Sgk1*) were significantly hypermethylated. Furthermore, the 5′ UTR region of casitas-B-lineage lymphoma protein-b (*Cblb*) and the 1 to 5 kb region of SMAD family member 4 (*Smad4*) were found to be hypomethylated in the heat-inactivated SV-53 group compared to the control group. No significant DMRs were found in mice receiving live SV-53 and PCA compared to untreated mice.

## 4. Discussion

Probiotics, prebiotics, and symbiotic products have been shown to promote a balanced gut microbiota, reduce inflammation, and enhance immune system function—not only gut immunity but also systemic and distant tissue immunity through the common mucosal immune system [1,34,35,36,37]. In this study, the immunomodulatory and anti-inflammatory properties of the novel probiotic SV-53 and prebiotic PCA were explored. Certain probiotic bacteria and prebiotics, including small polyphenol oligomers, act as antagonist ligands for Toll-like receptors (TLRs), effectively inhibiting associated inflammatory pathways [38,39]. SV-53 is a Gram-negative bacteria with probiotic characteristics [26]. The subtle distinction between bacterial lipopolysaccharides (LPS) derived from pathogenic Gram-negative bacteria and Gram-negative commensal bacteria in gut microbiota is responsible for the immunoinhibitory activity of Gram-negative commensals [40]. On the other hand, PCA is a metabolite product resulting from the fermentation of blueberry juice by SV-53 and deriving from complex polyphenols and quercetin [38], which is a TLR4 antagonist [41]. 

The results of the current study revealed that SV-53 and PCA administration could increase the population of IgA-producing B cells in the ileum, and SV-53 intake increased serum IgA concentration without increasing IgG levels. Within the intestinal environment, IgA-positive (IgA^+^) cells and secretory IgA (sIgA) play a vital role in maintaining intestinal mucosal immunity and homeostasis [42]. sIgA is the most abundant antibody class in the mucosal immune system and gut lumen [43]. It enhances gut mucosal immunity and homeostasis through various mechanisms, including quenching microbial components, neutralizing bacterial toxins, influencing gut microbial communities, enhancing antigens transport to immune cells in GALT, and downregulating proinflammatory *responses* [44]. On the other hand, IgG antibodies are potent effector molecules that activate innate immune cells and induce inflammation [45,46]. An increased IgG level is associated with intestinal inflammation and persistent immunopathology in the intestinal mucosa [47]. In addition, an increase in local commensal-specific IgG during intestinal inflammatory diseases, such as colitis, has been reported by Castro-Dopico et al. (2019), and this increase was associated with the magnitude of intestinal inflammation [48]. They also found that IgG immune complexes-stimulated colonic macrophages demonstrated a Th17-polarizing phenotype [48]. No changes in IgG levels were observed in response to probiotic intake, reflecting the effectiveness of the treatment in terms of improving gut mucosal immunity by producing sIgA without increasing IgG levels and inducing inflammation.

In addition, probiotic and prebiotic intake significantly decreased the level of IL-17A, the main proinflammatory cytokine secreted by Th17 cells [7], and levels of IL-6 and IL-23, major cytokines inducing pathogenic Th17 cells production and function, in the ileum tissues of mice [9,10]. IL-17A is involved in the pathogenicity of inflammatory disorders [10]. For instance, the involvement of IL-17A/IL-23 in the pathogenesis of IBD has been shown in both human and animal studies [49]. We have previously shown that the fermentation of blueberry juice with SV-53 significantly increases the quantity of polyphenols naturally present in the juice [50]. This novel product was found to inhibit IL-6/STAT3 signaling [51]. IL-6/STAT3/RORγt pathway induces Th17 cells differentiation [52], and IL-23/IL-23 receptor signaling phosphorylates the STAT3 and increases the expression of inflammatory cytokines such as IL-17A, leading to intestinal inflammation [53]. Moreover, increased IL-6 negatively affects gut mucosal immunity and leads to gut barrier dysfunction by damaging tight junctions [54,55]. Our results may provide some evidence regarding the potential role of SV-53 and PCA in modulating the IL6/STAT3 pathway. However, further studies are required to explore the role of SV-53 and PCA in regulating the STAT3 pathway.

Furthermore, we found an increase in the IL-10 concentration in the PCA group and an increase in the number of IL-10-producing cells in the SV-53 and PCA groups. IL-10 is an immunosuppressive cytokine secreted by various immune cells, including dendritic cells, macrophages, and Tregs. IL-10-producing CD4^+^ CD25^+^ Treg cells play a key role in controlling intestinal inflammation and self-tolerance [56]. IL-10 contributes to B cells differentiation and the production of IgA by B cells, thereby enhancing mucosal humoral immunity [57]. In addition, in a study using a chronic colitis mice model, IL-10 suppressed the inflammasome/IL-1β pathway, reduced the pathogenicity of Th17 cells, and suppressed IL-17 production, leading to the mitigation of intestinal inflammation [58]. Lactic acid bacteria have been shown to prevent intestinal inflammation and autoimmunity by increasing IL-10 and CD4^+^ CD25^+^ Treg cells [59,60]. 

Moreover, studies have shown that prebiotics have the ability to modulate IgA, IgG, proinflammatory, and anti-inflammatory cytokines [61]. For example, prebiotic intake increased sIgA and IL-10 levels in the intestines of rats [62] and inhibited LPS-induced IL-17 release in mouse splenocytes [63]. Additionally, polyphenols inhibit inflammation by blocking TLR4 and suppressing the production of inflammatory mediators such as IL-1β, IL-6, and TNFα [61].

Gut miRNAs link the gut immune system and the microbial community by regulating signaling pathways critical for maintaining gut microenvironment homeostasis [64]. In the current study, the effect of nutritional intervention with probiotic SV-53 and prebiotic PCA intake on miR-223 and miR-425 expression was explored, and a significant decrease in the relative expression of both miRNAs was observed in the treatment groups. A marked increase in the expression of miR-223 and miR-425 has been reported in intestinal inflammatory conditions such as IBD [65,66]. miR-223 may promote the secretion of cytokines by dendritic cells and therefore regulate Th17 and Tregs differentiation from naïve T cells [64]. miR-233 is downstream of the IL-23 cascade [67] and has been shown to promote pathological Th17 cells differentiation during autoimmunity [68]. miR-223 also increases intestinal barrier permeability by targeting thigh junction proteins by mediating the IL23/Th17 pathway [67]. Impaired barrier function is a major factor contributing to intestinal inflammation [67]. miR-425 is involved in chronic–degenerative inflammations such as autoimmune diseases, age-related disorders, and cancer progression [69]. This miRNA promotes Th17 cells differentiation and pathogenicity. In a study conducted on colitic mice, overexpression of miR-425 was found to mediate Th17 cell production and IL-17A secretion, while inhibiting miR-425 alleviated intestinal mucosal inflammation and markedly decreased IL-17A levels [65]. Therefore, these results suggest that SV-53 and PCA may enhance gut immunity and prevent inflammation by regulating gut miRNAs as well.

Similar to these results, several studies have reported the beneficial effects of probiotics in mitigating gut inflammation through the modulation of miRNA expression. For instance, the oral administration of two probiotics, *Lactobacillus fermentum* and *Saccharomyces boulardii,* significantly regulated the expressions of miR-223 and improved gut barrier function, restored dysbiosis, and ameliorated disease severity in a mouse model of dextran sodium sulfate (DSS)-induced colitis [70,71]. In another study, feeding mice with probiotic *E*. *coli* Nissle 1917, a Gram-negative probiotic bacterium, prevented the DSS-induced colonic inflammation by enhancing the expression of various cytokines and proteins responsible for maintaining epithelial integrity and decreasing several miRNAs involved in the inflammatory response, including miR-223 [66]. 

Full genome analysis of SV-53 has revealed a cluster of genes coding for bacteriocins against pathogenic Gram-negative bacteria [72]. We have previously shown the antibacterial effect of SV-53 against *Staphylococcus aureus* and *Salmonella enterica* serovar Typhimurium [26]. In this study, the microbiome analysis signified a decrease in *E. coli* population in mice fed SV-53 and the mixture of SV-53 and PCA. An increase in pathogenic *E. coli* has been observed in IBD and following antibiotic-induced dysbiosis [73,74]. Furthermore, studies have observed that the Western diet can alter gut microbiota composition, increase gut permeability, induce overgrowth of *E. coli,* promote invasive *E. coli* colonization in the gut mucosa, and subsequently lead to inflammation and immune dysfunction [75,76]. Moreover, enterotoxigenic *E. coli* exposure has been shown to increase RORγt expression and IL-17 secretion in the intestines of mice [77]. Several probiotics, such as *Lactobacillus plantarum* and *E. coli* Nissle 1917, have been reported to alleviate pathogenic *E. coli*-induced intestinal barrier dysfunction by modulating the expression of tight junction proteins [78]. 

Furthermore, we demonstrated the immunomodulatory and anti-inflammatory properties of heat-inactivated SV-53 by increasing IgA concentration in the serum and reducing IL-6 and IL-23 concentrations and the number of IL-17A and IL-6 producing cells in the ileum tissues of mice. However, our results indicated that viable SV-53 exhibited more robust immunomodulatory properties compared to the non-viable bacterium. Many of the ascribed benefits of live probiotics are facilitated by interactions of these organisms with the host’s gut epithelium and immune system. They can interact with the host directly through cell-to-cell interaction or indirectly through the production of metabolites and the release of various components [79]. Heat-inactive probiotics are also metabolically active and can interact with the host, resulting in health benefits [79]. In fact, heat-killed probiotics retain functionality due to the presence of cell wall components involved in interactions with the host, such as lipoteichoic acids and peptidoglycan [79]. 

In an experimental colitis model, the administration of a mixture of heat-inactivated probiotics downregulated IL-6, IL-23, STAT3, and p-STAT3 expression in colonic tissues of colitic rats [80]. In another study, heat-inactivated *E. coli* Nissle 1917 induced the antimicrobial peptide β-defensin in intestinal epithelial cells [81]. We have previously shown the immunomodulatory and anti-inflammatory activity of heat-inactivated lactic acid bacteria using primary cultures of intestinal epithelial cells in mice fed *Lactobacillus casei CRL* 431 or *Lactobacillus helveticus* R389 [82]. Similar to this study, Pyclik et al. demonstrated the significance of the viability status of *Bifidobacterium* strains for its immunomodulatory properties, with heat inactivation being shown to alter these properties [83]. In addition, Sturm et al. reported that in contrast to heat-killed *E. coli* Nissle 1917, the viable bacterium might exert a more potent inhibitory effect against intestinal inflammation by inhibiting T cells proliferation and the expression of proinflammatory cytokines [84].

To investigate whether the immunomodulatory properties of SV-53 are mediated, in part, through epigenetic modification, DNA methylation analysis was conducted. In mice exposed to heat-inactivated SV-53, an increase in CPGs methylation around TSS of several genes controlling Th17 differentiation and function, including *Il6*, *Il17rc*, *Il9*, *Il11*, *Akt1*, *Ikbkg*, and *Sgk1*, was found, which may indicate transcriptional repression of these genes.

As discussed earlier, IL-6 plays a central role in initiating the differentiation of naïve T cells into the Th17 lineage. IL-17RC is critical to signal transduction via IL-17A and the pathogenesis of autoimmune diseases. In vitro studies have shown that IL-17RC deficiency protects against autoimmunity [85]. Moreover, it has been shown that Th17 cells can secrete IL-9. IL-9 neutralization and IL-9 receptor deficiency led to a reduction in Th17 cells and IL–6-producing macrophages in the central nervous system, ultimately ameliorating the development of experimental autoimmune encephalomyelitis in mice [86]. IL-11 may induce Th17 cells generation and expansion in vitro and Th17 cells responses in vivo and contribute to autoimmunity. IL-11 exerts its effect on Th17 cells by activating the IL-6/STAT3 pathway [87].

Furthermore, Akt plays an important role in Th17 production. Upon TCR stimulation, the PI3k/AKT pathway contributes to the proliferation and survival of Th17 cells by facilitating nuclear transportation of RORγ and downregulating negative regulators of Th17 differentiation [88]. *Ikbkg*, also known as nuclear factor-kappaB (NF-κB) essential modulator (NEMO) encodes the regulatory subunit of the inhibitor of kappaB (IkB) kinase (IKK) complex, which is essential for NF-κB activation [89]. NF-κB contributes to the production of inflammatory cytokines such as IL-1, IL-6, IL-23, RORγ expression, and the generation of inflammatory T cells, including Th17 cells [90]. *Sgk1* encodes SGK1, a serine/threonine kinase, which is downstream of IL-23 signaling. *Sgk1* expression is specifically induced and maintained by exposure to IL-23. The kinase activity of SGK1 is higher in Th17 cells compared to other subsets of T cells. On the other hand, SGK1 is crucial for regulating IL-23R expression, thereby maintaining the Th17 cell phenotype by suppressing FoxO1, which negatively regulates IL-23R expression [91]. 

In addition, significant hypomethylation was detected within CPGs around TSS sites of *Cblb* and *Smad4* in heat-inactivated SV-53 mice compared to their untreated counterparts, which may suggest the transcriptional activation of these genes. Cblb is a Cbl family E3 ubiquitin ligase that restrains pathogenic Th17 cells generation and Th17-related autoimmunity by suppressing macrophages’ IL-6 secretion [92]. IL-21 promotes Th17 cells differentiation and SMAD4 negatively regulates IL-21-induced Th17 through repression of *Rorc* gene expression [93]. However, we could not find any changes in the methylation status of genes in mice receiving live bacteria, which indicates that in addition to cell wall components, other components released during the heat-inactivation process may contribute to epigenetic modification by heat-inactivated SV-53.

While results from cytokine and DNA methylation analyses may offer some insights into the potential role of SV-53 in regulating Th17 cells, flow cytometry analysis needs to be performed to elucidate the effect of treatment on Th17 cells production.

In conclusion, the results of this study suggest the potential role of both viable and heat-inactivated probiotic SV-53, along with the prebiotic PCA, in improving gut immunity and homeostasis by modulating IgA, cytokines, and/or gut miRNAs expression. Additionally, these findings highlight the significant role of heat-inactivated SV-53 in regulating gut immunity through DNA methylation. SV-53 also enhances gut immunity and prevents inflammation by decreasing the population of pathogenic bacteria. These findings may indicate the probiotic characteristics of SV-53 as a novel next-generation probiotic bacterium.

## Figures and Tables

**Figure 1 microorganisms-11-02456-f001:**
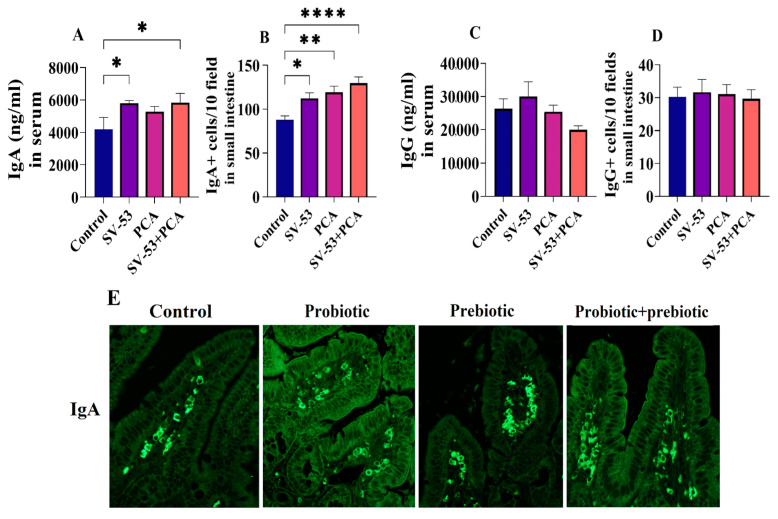
The effect of the probiotic and prebiotic intake on gut mucosal immunity. Female Balb/c mice were treated with the SV-53, PCA, or SV-53 + PCA mixture administered in their drinking water for three weeks. Serum levels of IgA and IgG and the number of IgA^+^ and IgG^+^ B cells in the ileum tissues of mice were measured via ELISA and direct immunofluorescence, respectively; (**A**) the serum concentration of IgA, (**B**) the number of IgA^+^ B cells, (**C**) the serum concentration of IgG, and (**D**) the number of IgG^+^ B cells. One-way ANOVA followed by Dunnett’s multiple comparisons were used to compare groups. All values are mean ± SEM. N = 6, * *p* < 0.05, ** *p* < 0.01 and **** *p* < 0.0001 vs. control. (**E**) Immunofluorescence images of histological sections of the ileum stained with the appropriate dilution of anti-IgA antibody (1:100) and imaged using a fluorescent light microscope at 40× magnification.

**Figure 2 microorganisms-11-02456-f002:**
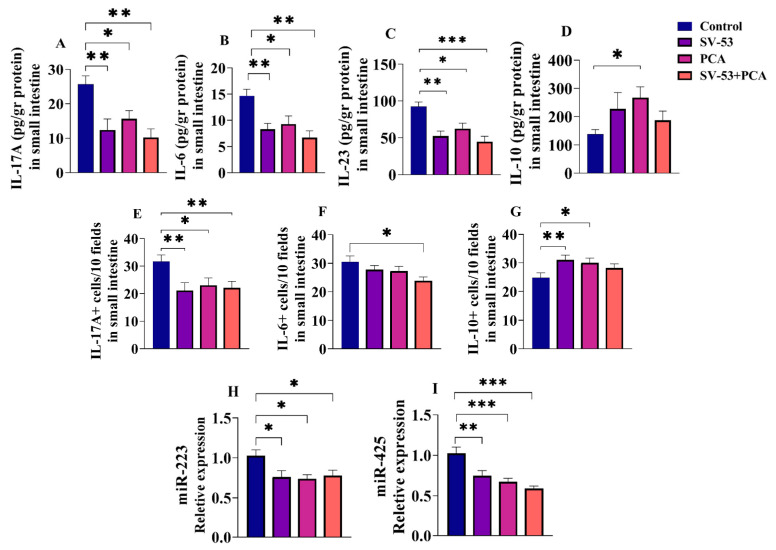
The effect of the probiotic and prebiotic intake on selected cytokines and miRNAs in the ileum tissues of mice. Female Balb/c mice were treated with the SV-53, PCA, or SV-53 + PCA mixture in drinking water for three weeks. The concentrations of (**A**) IL-17A, (**B**) IL-6, (**C**) IL-23, and (**D**) IL-10 in the ileum tissues of mice were measured via ELISA. The frequencies of (**E**) IL-17A^+^, (**F**) IL-6^+^, and (**G**) IL-10^+^ cells were determined via immunofluorescence. The expression of (**H**) miR-223 and (**I**) miR-425 was measured via RT-qPCR. One-way ANOVA followed by Dunnett’s multiple comparisons was used to compare groups. All values are mean ± SEM. N = 6, * *p* < 0.05, ** *p* < 0.01, and *** *p* < 0.001 vs. control.

**Figure 3 microorganisms-11-02456-f003:**
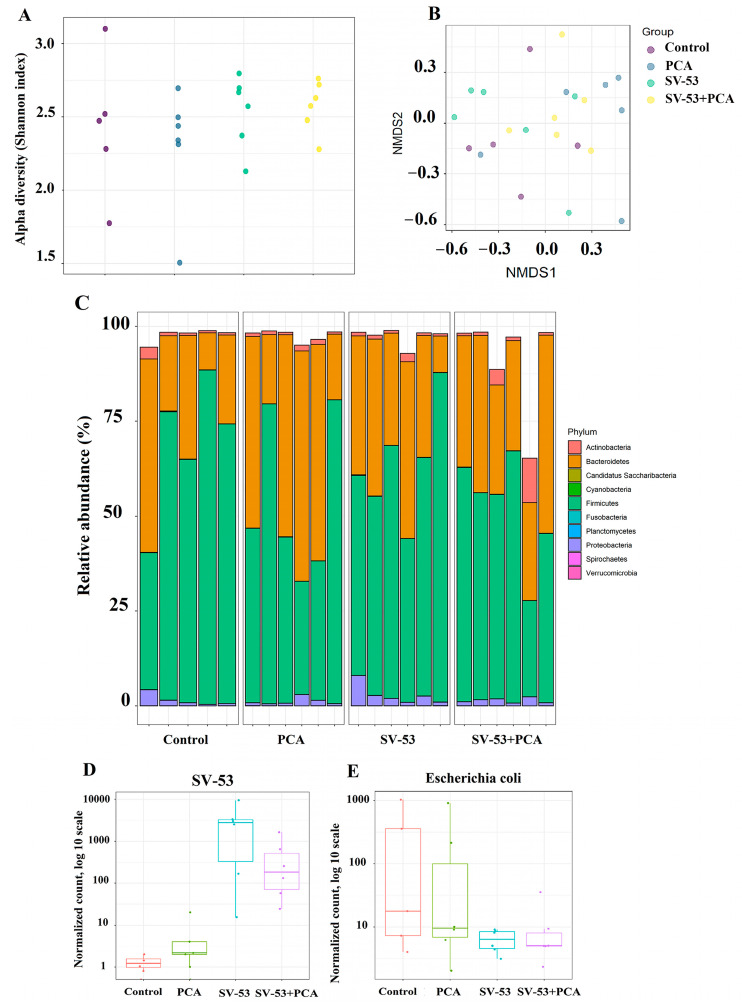
The effect of the probiotic and prebiotic intake on the gut microbiome. Female Balb/c mice were treated with the SV-53, PCA, or SV-53 + PCA mixture in drinking water for three weeks. Then, cecum contents of mice were used to analyze the gut microbiome via shallow shotgun sequencing; (**A**) alpha diversity, (**B**) beta diversity, (**C**) bacterial composition at the phylum level (ten most abundant phyla), and (**D**,**E**) differentially abundant species. n = 5 in the control group and n = 6 in the treatment groups.

**Figure 4 microorganisms-11-02456-f004:**
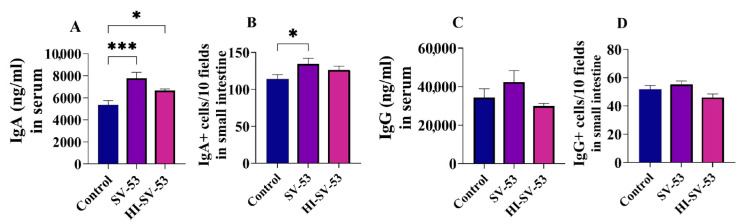
The effect of the heat-inactivated probiotic intake on gut mucosal immunity. Female Balb/c mice were treated with live SV-53 or heat-inactivated SV-53 (HI-SV-53) in drinking water for three weeks. Then, the serum levels of IgA and IgG, and the number of IgA^+^ and IgG^+^ B cells in the ileum tissues of mice were measured by ELISA and immunofluorescence, respectively; (**A**) the concentrations of IgA, (**B**) the number of IgA^+^ B cells, (**C**) the concentrations of IgG, and (**D**) the number of IgG^+^ B cells. One-way ANOVA followed by Dunnett’s multiple comparisons were used to compare groups. All values are mean ± SEM. N = 6, * *p* < 0.05 and *** *p* < 0.001 vs. control.

**Figure 5 microorganisms-11-02456-f005:**
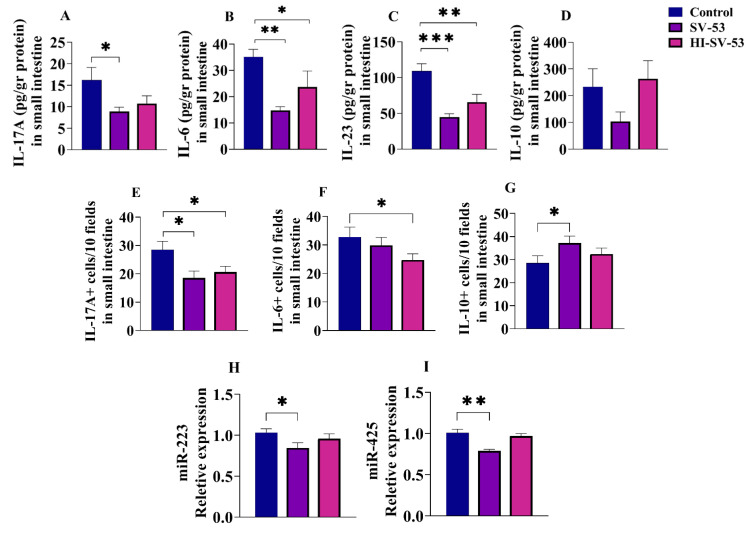
The effect of the heat-inactivated probiotic on selected cytokines and miRNAs expression in the ileum tissues of mice. Female Balb/c mice were treated with live SV-53 or heat-inactivated SV-53 (HI-SV-53) in drinking water for three weeks. The concentrations of (**A**) IL-17A, (**B**) IL-6, (**C**) IL-23, and (**D**) IL-10 in the ileum tissues of mice were measured via ELISA. The frequencies of (**E**) IL-17A^+^, (**F**) IL-6^+^, and (**G**) IL-10^+^ cells were determined via immunofluorescence. The expression of (**H**) miR-223 and (**I**) miR-425 were measured via RT-qPCR. One-way ANOVA followed by Dunnett’s multiple comparisons was used to compare groups. All values are mean ± SEM. N = 6, * *p* < 0.05, ** *p* < 0.01 and *** *p* < 0.001 vs. control.

**Figure 6 microorganisms-11-02456-f006:**
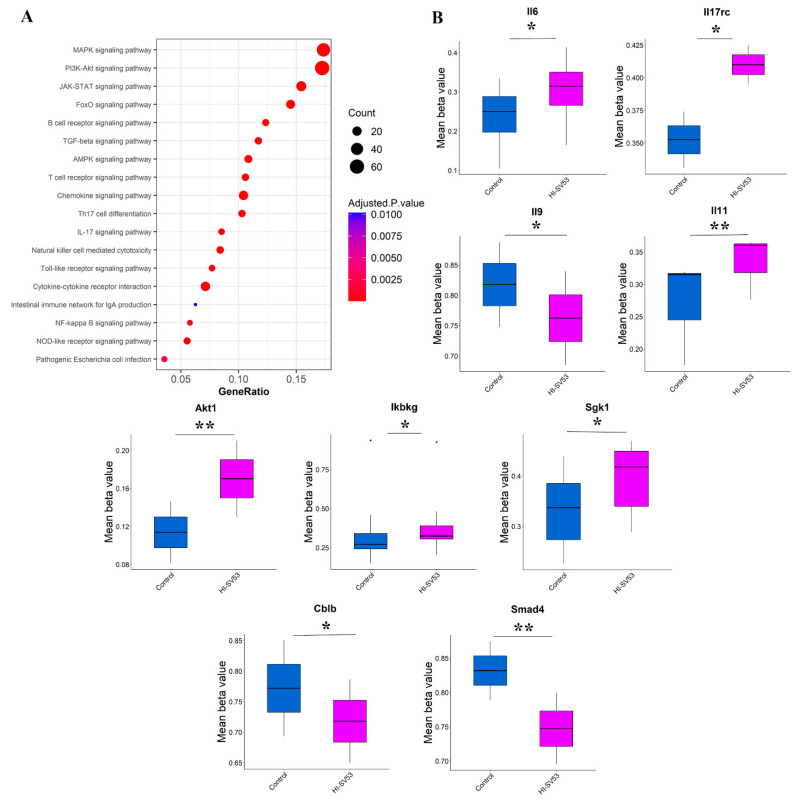
The effect of the heat-inactivated probiotic on DNA methylation. Female Balb/c mice were treated with live SV-53 or heat-inactivated SV-53 (HI-SV-53) administered via drinking water for three weeks. (**A**) Pathways enrichment analysis visualized by Enrichr, (**B**) boxplots of differentially methylated genes related to Th17 cells differentiation and function. * *p* < 0.05 and ** *p* < 0.01 vs. control.

## Data Availability

The data presented in this study are available on request from the corresponding author.

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
