# Peer review of "Novel Probiotic Bacterium Rouxiella badensis subsp. acadiensis (Canan SV-53) Modulates Gut Immunity through Epigenetic Mechanisms"

_microorganisms, 2023, doi:10.3390/microorganisms11102456_

Round 1

Reviewer 1 Report

The chance to examine the paper titled " Novel Probiotic Bacterium Rouxiella badensis subsp. acadiensis (Canan SV-53) Modulates Gut Immunity through Epigenetic Mechanisms" made me extremely happy. This work examined the immunomodulatory and anti-inflammatory properties of live, heat-inactivated forms of Rouxiella badensis subsp. acadiensis (Canan 21 SV-53) as well as its metabolite protocatechuic acid. It might be beneficial for the possible use of the novel probiotic bacterium SV-53 and prebiotic PCA in enhancing gut immunity and homeostasis. Overall, it is a finished product that should be published in the journal. The suggestions for specifics were as follows:

1 Line 306-309, Page 8

Only the abundance of SV-53 and E. coli was examined during the examination of the microbiome; if there were any beneficial bacteria with increasing abundance, it is suggested to increase the analysis results to better demonstrate the improving gut homeostasis. 

2 Line 388-389, Page 11

It seems from the previous data that live SV-53 bacteria are more efficient than heat inactivated Sv-53 bacteria in increasing IgA levels and the selected cytokines and miRNAs. Only heat-inactivated SV-53, however, demonstrated a significant difference from the control in DNA methylation analysis; please include a justification for this finding in the discussion area.

Author Response

Dear Reviewer,

Thank you so much for your valuable comments.

Reviewer 2 Report

Reviewer comments

 Dear Chantal Matar

corresponding author

Microorganisms

Manuscript ID: microorganisms-2641688

Title:

“Novel Probiotic Bacterium Rouxiella badensis subsp. acadiensis 2 (Canan SV-53) Modulates Gut Immunity through Epigenetic 3 Mechanisms”

Major comments

Abstract

Authors should clear the results statements in the abstract.

Materials and Methods

“2-probiotic group; receiving probiotic SV-53 (5 × 107 CFU/mL)”

The authors should clarify how they ensure this strain of bacteria from the previous preparations. Authors should mention this in the preparation section.

Authors should provide the dose volume of probiotics.

2.3. Study Design

The design should be rearranged into five groups: 1. Control, 2. probiotic (live SV-35), 3.  prebiotic, 4. mixed group, 5. heat-inactivated bacteria.

2.2. Probiotic and Prebiotic Solution Preparation

Please clarify the preparation of the probiotic solution

2.6. Determination of IgA, IgG, IL-17A, IL-6, IL-23, and IL-10 Concentrations by ELISA

“The concentration of IgA, IgG, IL-17A, IL-6, IL-23, and IL-10 was determined in serum and/or ileum tissues.”

These concentrations were determined in tissues and mentioned in the previous section. Please delete “ileum tissues” from the statement.

Results

“In the intestine, IgA-positive 241 (IgA+) cells and secretory IgA (sIgA) exert a vital role in maintaining intestinal mucosal 242 immunity and homeostasis [33], while increased IgG level is associated with intestinal 243 inflammation and persistent immunopathology in the intestinal mucosa [34]. “Please, transfer to the discussion section.

- Please provide one term throughout the manuscript, such as small intestine and ileum.

- The authors repeat the statements of materials in the results. Please delete repeated statements.

L 332: “serum IgA levels of IgA (p<0.05)” rewrite

L406: please write the full name of “Bacterial lipopolysaccharides (LPS)” 

What mechanisms? The authors should debate the mechanism. 

An obvious critical comment is the poor English language level in terms of syntax, punctuation, and grammatical errors (for example, put articles (a, an, and the) correctly. A native English speaker should edit the manuscript text carefully.

Since English is not our mother tongue, it would be better if the authors could check the manuscript again to remove some typos and syntax.

Author Response

(The authors gave the same response as above.)
